# *Leuconostoc citreum* Inhibits Adipogenesis and Lipogenesis by Inhibiting p38 MAPK/Erk 44/42 and Stimulating AMPKα Signaling Pathways

**DOI:** 10.3390/ijms24087367

**Published:** 2023-04-17

**Authors:** Hyo-Shim Han, Ilavenil Soundharrajan, Mariadhas Valan Arasu, Dahye Kim, Ki-Choon Choi

**Affiliations:** 1Department of Biotechnology, Sunchon University, Suncheon 57922, Republic of Korea; kkruki@hanmail.net; 2Grassland and Forages Division, National Institute of Animal Science, Rural Development Administration, Cheonan 31000, Republic of Korea; ilavenil@korea.kr; 3Department of Botany and Microbiology, College of Science, King Saud University, P.O. Box 2455, Riyadh 11451, Saudi Arabia; mvalanarasu@gmail.com; 4Animal Genomics and Bioinformatics Division, National Institute of Animal Science, Jeonju 55365, Republic of Korea

**Keywords:** *Leuconostoc citreum*, LSC, 3T3-L1, adipogenesis, lipogenesis, lipolysis

## Abstract

Probiotics provide a range of health benefits. Several studies have shown that using probiotics in obesity treatment can reduce bodyweight. However, such treatments are still restricted. *Leuconostoc citreum*, an epiphytic bacterium, is widely used in a variety of biological applications. However, few studies have investigated the role of *Leuconostoc* spp. in adipocyte differentiation and its molecular mechanisms. Therefore, the objective of this study was to determine the effects of cell-free metabolites of *L. citreum* (LSC) on adipogenesis, lipogenesis, and lipolysis in 3T3-L1 adipocytes. The results showed that LSC treatment reduced the accumulation of lipid droplets and expression levels of CCAAT/ enhancer-binding protein-α & β (C/EBP-α & β), peroxisome proliferator-activated receptor-γ (PPAR-γ), serum regulatory binding protein-1c (SREBP-1c), adipocyte fatty acid binding protein (aP2), fatty acid synthase (FAS), acetyl CoA carboxylase (ACC), resistin, pp38MAPK, and pErk 44/42. However, compared to control cells, adiponectin, an insulin sensitizer, was elevated in adipocytes treated with LSC. In addition, LSC treatment increased lipolysis by increasing pAMPK-α and suppressing FAS, ACC, and PPAR-γ expression, similarly to the effects of AICAR, an AMPK agonist. In conclusion, *L. citreum* is a novel probiotic strain that can be used to treat obesity and its associated metabolic disorders.

## 1. Introduction

Globally, obesity is becoming an epidemic issue, with an increasing occurrence rate. Obesity rate is 30.4% in the United States, 12.8% in Europe, and 10.7% in China [1,2,3]. Global assessments have shown that almost 2.3 billion children and adults are overweight or obese. If the current conditions continue, 2.7 billion adults might be overweight or obesogenic in 2025. The World Health Organization (WHO) forecasts that 39% of people in 2035 might become obese [4]. According to Organization for Economic Co-operation & Development (OECD), more than 4% and almost 30% of the adult population in Korea are obese and overweight, respectively. Jung et al. 2020 projected that Korean adults would have a median body mass index (BMI) of 23.55 kg/m^2^ in 2040. Based on BMI classification, 70.05% of all adults will become obese by 2040 [5]. As obesity increases, it poses a greater public health threat and economic problem in the long run because it is closely linked to several chronic diseases, including cardiac disease, aging, cancer, diabetes mellitus, skeletal muscle illness, inflammatory diseases, and fatty liver accumulation [6,7]. Every year, obesity and its related diseases kill more than 2.8 million people in worldwide [8]. Although several factors, including a sedentary lifestyle, high calorie intake, depression, social and monetary issues, contribute to obesity, one common cause is the accumulation of lipids in white adipocytes through adipogenesis and lipogenesis [9]. The mechanism of depositing lipids in adipocytes has several complex processes involving several genes and transcriptional factors. Among various factors, CCAAT/enhancer-binding protein (C/EBPs) and peroxisome proliferator-activated receptor-γ (PPAR-γ) are key transcriptional genes that can induce lipogenesis-associated genes such as adipocyte fatty acid binding protein (aP2), fatty acid synthase (FAS), acetyl CoA carboxylase (ACC) [10,11,12]. Thus, most researchers are focused on developing dietary supplements to control excessive fat deposition in adipocytes. In terms of anti-obesity effectiveness, probiotics are among the major factors [13,14,15]. It has been shown that probiotics have anti-obesity effects by altering metabolic energy, improving the intestinal barrier, increasing metabolism, improving immune response, modulating nerve activity, and modulating appetite [15,16,17].

In the modern era, many drugs are available for treating obesity and its associated disorders. However, they can cause some adverse effects such as nausea, insomnia, constipation, gastrointestinal problems, and cardiovascular problems [18]. A potent strategy is required to find and develop anti-obesity supplements that can reduce fat deposition in the body, reduce the risk of obesity-related diseases, and minimize side effects. Recently, lactic acid bacteria have received a considerable amount of attention due to their effects on obesity and its associated metabolic disorders [19,20,21,22,23,24]. Several studies have shown that *Leuconostoc* spp.-mediated food supplements reduce obesity and metabolic diseases associated with obesity [25,26,27]. There has been a substantial variance in the present study in comparison to previously reported data on *Leuconostoc* species. We used a novel strain of *Leuconostoc citreum* in the present study to determine its efficiency as regards inhibiting differentiation and lipid accumulation in 3T3-L1 adipocytes. Therefore, the *L. citreum* strain was cultured in a 10% FBS-DMEM (Fetal Bovine Serum-Dulbecco’s Modified Eagle Medium) medium for the production of secondary metabolites. The secondary metabolites in the sample were then lyophilized. The effects of the cell-free supernatant of *L. citreum* (LSC) on the differentiation of adipocytes and fat deposition have been studied. A further investigation was conducted on the molecular mechanisms that may underlie LSC’s inhibition of lipogenesis and lipolysis.

## 2. Results

### 2.1. Impact of Cell Free Supernatant of L. citreum (LSC) on Cell Viability

As shown in Figure 1A,B, the effects of LSC at different concentrations (0.5 mg/mL to 0.001 mg/mL) on viability of 3T3-L1 adipocytes were observed at 24 h and 48 h. As compared to control cells, cells treated with LSC at a concentration of 0.5 mg/mL had a significantly lower viability at 24 h and 48 h. A concentration of LSC of less than 0.250 mg/mL did not affect cell morphology or viability, suggesting that LCS could be used for further experimental analysis.

### 2.2. LSC Treatment Reduces Fat Deposition in 3T3-L1 Cells

We first examined the effects of LSC at concentrations of 0.05, 0.1, and 0.15 mg/mL on differentiation and fat deposition on day 10 by Oil Red O stain (ORO). Microscopic examination revealed higher fat deposition spots at several locations with large sizes in control cells. The results showed that insulin, IBMX, and dexamethasone successfully induced differentiation and fat deposition in adipocytes. In contrast, cells treated with LSC at different concentrations from the beginning of differentiation to the end of the experiment showed reduced fat deposition spots and fat size in a dose-dependent manner. At 0.15 mg/mL, fat deposition was strongly inhibited compared to other dose ranges. LSC at 0.1 mg/mL also strongly decreased fat deposition in cells during differentiation (Figure 2). Additionally, the ORO strain was extracted from experimental adipocytes using 99% 2-propanol and absorbance was measured at 450 nm, providing strong evidence for microscopic visualization. The percentage of lipid deposition in experimental cells revealed that LSC treatment reduced lipid content in a dose-dependent manner compared to control on day 10. LSC at 0.15 mg/mL inhibited the most fat deposition (Figure 2).

### 2.3. LSC Downregulates Differentiation and Fatty Acid Synthesis Associated Proteins

Microscopical observations and lipid quantification data confirmed that LSC treatment reduced adipocyte fat accumulation. Western blot was used to determine molecular mechanisms involved in LSC treatment-induced fat reduction in adipocytes. Key transcriptional factors such as PPAR-γ, C/EBP-α/β, and SREBP-1c associated with differentiation were downregulated in adipocytes treated with LSC on day 10 (Figure 3A). LSC treatment also inhibited the expression of key lipogenesis-associated proteins such as FAS, ACC, and aP2. Other proteins related to insulin resistance and insulin sensitivity such as resistin and adiponectin levels were also determined. Results showed that LSC treatment decreased resistin but increased adiponectin expression in adipocytes compared to control (Figure 3A,B).

### 2.4. LSC Competes with Rosiglitazone (RGZ)-Induced Lipid Accumulation and PPAR-γ Expressions

The effect of LSC treatment on fat deposition can be attributed to the downregulation of key adipogenesis and lipogenesis-associated proteins. The effects of rosiglitazone (RGZ), a PPAR-γ agonist, and LSC on PPAR-γ and fat accumulation in adipocytes were then determined. RGZ treatment alone rapidly increased fat deposition in adipocytes by upregulating PPAR-γ expression compared to the control, whereas treatment with LSC alone strongly decreased fat accumulation by downregulating PPAR-γ expression compared to the control (Figure 4A–C). Cells treated with LSC in the presence of RGZ significantly reduced the lipid content of cells and PPAR-γ expression compared to RGZ alone. However, there was no statistically significant difference between the control and the LSC + RGZ treatment.

### 2.5. LSC Inhibits Differentiation and Fat Deposition through p38MAPK and Erk1/2 Signaling Pathways

LSC treatment significantly inhibited fat accumulation in adipocytes through adipogenesis and lipogenesis-related proteins. The effects of LSC on signaling pathways related to adipocyte differentiation and lipid accumulation were then investigated. Western blot analysis revealed that LSC treatment inhibited phosphorylation of p38MAPK at Thr180/Tyr182 and Erk1/2 at Thr202/Tyr204 as compared with the control (Figure 5).

### 2.6. LSC Induces Lipolysis by Activating AMPK-α in Differentiated Adipocytes

Differentiated adipocytes treated with LSC at 0.15 mg/mL or AICAR (5-Aminoimidazole-4-carboxamide ribonucleotide) at 1 mM, an AMPK-α agonist, for 12 h increased phosphorylation of AMPK-α closely associated with lipolysis but decreased PPAR-γ, FAS, and ACC expression levels closely associated with adipocyte differentiation and fatty acid synthesis (Figure 6). The results obtained for LSC were significantly comparable with those obtained for the AICAR treatment.

## 3. Discussion

In this study, we investigated the anti-adipogenic and anti-lipidemic properties of cell-free supernatants produced by *Leuconostoc citreum* (LSC) in adipocytes. The number of studies exploring the beneficial effects of the microbiome in humans has increased recently [28,29]. In general, probiotics can colonize epithelial cells of the gastrointestinal tract (GIT) and regulate key signaling molecules that are closely associated with human health [30]. In addition, post-biotics derived from gut-associated probiotics are essential candidates for the development of therapeutics against obesity [31]. Moreover, probiotics can produce many secondary metabolites and branched chain fatty acids that could reduce obesity [32]. Several researchers are actively involved in the production of fermented products in the presence of *Leuconostoc spp* with significant biological potential. As examples, the soymilk fermented with *L*. *kimchi, L. citreum* and *L. plantarum* significantly reduced fat deposition in 3T3-L1 adipocytes by inhibiting key transcription factors C/EBP-α and PPAR-γ. Furthermore, Soypro treatment reduced low density lipoprotein cholesterol (LDL) levels without affecting body weight in obese rats [25]. Another study reported that pear extract and robusta fermented with *L. *mesenteroids** significantly reduced body weight and adipose tissue mass, and the size of lipids in liver in obese rats compared to control rats [27,33]. Supplementation with *L. mesenteroids* reduced blood urea nitrogen, glucose, and triglyceride levels in obese mice serum, as well as fatty liver development and liver steatosis in comparison with controls [26,34]. Thus, we investigated the effects of LSC on fat deposition and differentiation in 3T3-L1 adipocytes. However, a significant difference exists between this study and previous studies on *Leuconostoc* species. The results of this study showed that the amount and size of fat deposition spots in cells treated with LSC at different concentrations were significantly reduced. In the presence of a differentiation-inducing cocktail, huge amounts of lipid accumulation with large sizes were observed in several places of adipocytes. However, LSC treatment reduced lipid accumulation in a dose-dependent manner. This study confirmed that LSC could exert anti-adipogenic and anti-lipogenic effects.

Adipogenesis is regulated by a variety of transcription factors, including C/EBP family members and PPAR-γ [35]. At an early stage of differentiation, C/EBP-β and C/EBP-δ become highly expressed in adipocytes. They can act as positive modulators of differentiation and lipid accumulation [11,36]. C/EBP-β is considered the most important factor induced by adipogenic cocktail stimuli. Knockdown of C/EBP-β can inhibit adipogenesis in 3T3-L1 cells [37,38,39]. PPAR-γ is the master regulator of differentiation of adipocytes and metabolism [10]. PPAR-γ and C/EBP-α cooperate with each other to orchestrate the entire adipogenic program [40]. In the present study, adipogenic stimulation significantly increased fat deposition in adipocytes, whereas LSC treatment significantly reduced fat accumulation compared to the control. The mechanisms behind the inhibition of differentiation and lipid accumulation in adipocytes in response to LSC treatment were determined. The results suggested that C/EBP-β was downregulated in adipocytes treated with different concentrations of LSC compared to the control cells. Subsequently, the expression levels of PPAR-γ and C/EBP-α protein also decreased following LSC treatment. A competitive study was performed between LSC and rosiglitazone (RGZ), an agonist for PPAR-γ. RGZ treatment increased fat deposition and PPAR-γ expression in adipocytes, whereas LSC treatment in the presence of RGZ significantly reduced fat accumulation as well as PPAR-γ expression compared to RGZ treatment. This study showed that LSC treatment could abolish RGZ-induced fat accumulation and expression of PPAR-γ.

Other transcriptional factors such as cyclic AMP response element (CREB) and sterol regulatory binding protein-1 (SREBP-1) can promote adipogenesis and differentiate pre-adipocytes into mature adipocytes. SREBP-1 is highly expressed in adipocytes [41,42]. It can stimulate PPAR-γ expression and fatty acid synthesis [11,43]. It also regulates genes responsible for lipogenesis such as FAS [44]. LSC significantly downregulated SREBP-1c during the differentiation of adipocytes. The downregulation of SREBP-1c is also one of the reasons for the inhibition of differentiation and the expression of PPAR-γ, suggesting that downregulation of C/EBP-β expression in LSC treated cells simultaneously reduced the expression of PPAR-γ, C/EBP-α, and SREBP-1c, which reduced adipocyte differentiation. An induction of PPAR-γ, C/EBP-α, and SREBP-1c can promote fatty acid synthesis by increasing key proteins associated with lipogenesis such as FAS and ACC [45,46]. In the terminal phase of differentiation, FAS, ACC, and aP2 expression levels are increased by several folds (>20 folds) [47,48]. ACC, FAS, and aP2 expression levels are positively associated with fatty acid synthesis. Among these enzymes, FAS and ACC are responsible for the synthesis of lipids and triglycerides. The key event in lipid metabolism is malonyl CoA production via carboxylation of acetyl CoA by acetyl CoA carboxylase. In the present study, we found that LSC significantly decreased the expression of FAS and ACC, which limited the synthesis of fatty acids in adipocytes, which was positively correlated with Oil Red O-stained lipid accumulation in LSC and control adipocytes. aP2 is only secreted by adipocytes. Several metabolic disorders and cardiovascular diseases are closely associated with high levels of aP2. Moreover, inhibiting aP2 might be a good strategy to reduce insulin resistance and type-2 diabetes [49,50]. Adipokines such as adiponectin, resistin, and leptin are physiological active cytokines. They are secreted from fat cells. They are well-known to influence adipogenesis [51,52,53]. High adiponectin levels in the circulation are associated with a lower incidence of diabetes [54,55]. However, a high level of resistin is associated with insulin resistance and diabetes [56,57]. In the present study, cells treated with LSC showed significantly increased adiponectin levels but reduced resistin protein expression in adipocytes. As a result of the current study, adipocyte differentiation was markedly influenced by LSC treatment, which reduced aP2 and resistin expression levels but increased adiponectin expression, suggesting that LSC treatments might negatively affect insulin resistance and diabetes development by improving insulin sensitivity and glucose uptake by cells.

Adipocyte differentiation is directed by p38MAPK and ERK1/2 [58]. By inhibiting p38MAPK and ERK1/2 phosphorylation, specific inhibitors have been shown to decrease adipocyte differentiation and lipid accumulation [59,60,61], suggesting that both p38MAPK and ERK1/2 initiations are crucial to lipogenesis and adipogenesis. Cells treated with LSC showed lower levels of phosphorylation of p38MAPK at Thr180/Tyr182 and Erk1/2 at Thr202/Tyr204 than control cells, which inhibits key transcriptional factors PPAR, C/EBP-α and C/EBP-β as well as lipogenesis-associated enzymes FAS and ACC [24]. AMPK can regulate fatty acid metabolism, thermogenesis, and adipose tissue development [62]. Activation of AMPK can inhibit lipogenesis and enhance fatty acid oxidation by inhibiting ACC and FAS and restoring carnitine palmitoyltransferase 1 (CPT1) [63,64]. Activated AMPK can inhibit adipocyte differentiation by inhibiting early mitotic clonal expansion phase, resulting in the reduced expression of adipogenic and lipogenic markers such as FAS, SREBP-1c, and aP2 [65,66]. In addition, activation of AMPK can reduce fat accumulation and the expression of PPAR-γ, C/EBPα, and early adipogenic transcriptional factors such as C/EBPβ and C/EBPδ [67]. Therefore, we determined the impact of LSC on lipolysis related molecular mechanism in differentiated adipocytes compared to the effect of AICAR, an AMPK agonist that could activate AMPK-α via phosphorylation and inhibit the expression of PPAR-γ, FAS, and ACC [68,69]. The results revealed that treatment with both LSC or AICAR for 12 h significantly increased the activated AMPK-α compared to control. This activation further downregulated the expression of adipogenic key transcriptional factor such as PPAR-γ and lipogenic enzymes such as FAS and ACC, suggesting that LSC could inhibit adipocyte differentiation and fat deposition by inhibiting adipogenic and lipogenic proteins while increasing lipolysis through the activation of AMPK-α by increasing its phosphorylating level. The results after LSC treatment were comparable to those after AICAR treatment. 

## 4. Materials and Methods

### 4.1. Isolation and Characterization of Leuconostoc citreum

De Man Rogosa and Sharpe Agar (MRS agar, CONDA, Madrid, Spain) medium was used for isolating *Leuconostoc citreum* (LSC) from whole crop rice samples. Biochemical analysis and 16SrRNA sequencing were used to identify the bacteria at the species level (Solgent Co, Seoul, Republic of Korea).

### 4.2. Production of Cell Free Supernatant of L. citreum (LSC) and Lyophilization

To obtain a fresh culture, LSC was grown in MRS broth and incubated at 37 °C for 48 h with mild shaking (125 rpm). LSC was prepared by centrifuging cultured LSC at 4000× *g* for 60 min at 4 °C, filtrating with filter membranes having different pour sizes, and then filtered sample was lyophilized at −40 °C under less than 50 m Torr pressure for 72 h (Ilshin Lab. Co., Ltd., Ansan-si, Gyeonggi-do, Republic of Korea) The BreeZe mini (sun clean bactericide, 30,000 ppm, Mirai Co., Chiba, Japan) was used to sterilize the LSC powder [24].

### 4.3. Cytotoxic Effects of LSC

Preadipocytes of 3T3-L1 (3T3-L1, ATCC-173, Manassas, VA, USA) were seeded into 96-well cell culture plates containing 10% fetal bovine serum in Dulbecco’s modified eagle medium (FBS-ATCC 30-2020 and DMEM ATCC 30-2002, Manassas, VA, USA) at a density of 10,000 cells/well. The plates were then incubated at 37 °C with 5% CO_2_ for 24 h. Different doses (0.5 mg–0.001 mg/mL) of LSC were added to the cells followed by incubation at 37 °C with 5% CO_2_ for 24 h and 48 h. EZ-cytox reagent (DoGenBio, Seoul, Republic of Korea) (10 µL/well) was added to each well followed by incubation at 37 °C with 5% CO_2_ for a further 30 min. Optical intensity was then measured with i3 Spectramax (Molecular Device, San Jose, CA, USA). The percentage of cell viability following LSC treatment compared to that of control cells was then calculated.

### 4.4. Differentiation and Lipid Deposition Induction in 3T3-L1 Cells

Cells were seeded into 12-well or 6-well plates at 15,000 or 30,000 adipocytes/well, respectively. Plates were incubated at 37 °C with 5% CO_2_. The incubations were continued for an additional 48 h in order to arrest growth. After 48 h, differentiation and lipid accumulation were initiated with 10% FBS-DMEM: 30–2002 medium containing insulin (1 μg/mL), IBMX (0.5 mM), and dexamethasone (1 mM). The cells were then switched to an insulin medium for 48 h. For treatment, cells were treated with different concentrations of cell free Supernatant of *L*. citreum (LSC) from the starting day of differentiation to the end of the experiment [70].

### 4.5. Detection and Determination of Fat Deposition by Oil Red O (ORO) Staining

Differentiated cells were fixed with formalin (10% in PBS) for an hour and then washed three times with PBS (Thermo Fisher Scientific, Waltham, MA, USA). Afterwards, cells were incubated with 60% isopropyl alcohol (IPA) for five minutes, covered with ORO (Sigma-Aldrich, St. Louis, MO, USA) working solution, and then incubated at room temperature for ten minutes. The ORO solution was discarded and the cells were washed 3–5 times with water or until excess stain was eliminated. ORO-stained cells were captured with an Evos cell image system (Fisher Scientific, Waltham, MA, USA). The ORO stain was then extracted from experimental cells and its intensity was measured at 490 nm. The percentage of fat deposition in cells treated with LSC compared with that in control cells was then calculated [24].

### 4.6. Proteins Extraction and Immunoblot Analysis

On day 10, the protein was extracted from experimental cells using RIPA cell lysis buffer containing 1× phosphatase and protease inhibitors (Roche, Basel, Switzerland; and Sigma-Aldrich, St. Louis, MO, USA). Protein content in cells was determined using the BCA method (Thermo Fisher Scientific, Waltham, MA, USA). An equal concentration of proteins from each experimental group was separated using mini SDS-PAGE gels (Biorad, Hercules, CA, USA). Afterwards, the proteins were transferred to polyvinylidene difluoride membranes (PVDF) using a semiwet transfer method (Turbo Transfer Gel method, Biorad, Hercules, CA, USA). Targeted proteins were immunoblotted overnight at 4 °C with primary antibodies (Cell Signaling Technology, Danvers, MA, USA). Membranes were then incubated with secondary antibodies linked to HRP (Cell Signaling Technology, Danvers, MA, USA). ECL reagent (Biorad, Hercules, CA, USA) was used to detect targeted protein bands. The intensity of the protein band was quantified using ImageJ software, 1.49 versions (Wayne Rasband, National institute of Health, Bethesda, ML, USA [19].

### 4.7. Statistical Analysis

Statistical analysis was performed for experimental data using SPSS16.0 (SPSS-Version 16.0, SPSS Inc., Chicago, IL, USA). One-way ANOVA and independent *t*-test were used to determine statistical significance between experimental samples at *p*-value less than 0.05.

## 5. Conclusions

Cell-free supernatants from *Leuconostoc citreum* (LSC) could inhibit adipogenesis and lipogenesis by downregulating the expression of main differentiation transcriptional factors PPAR-γ, C/EBP α, and C/EBP β and vital lipogenic enzymes FAS and ACC through the p38 MAPK and Erk 44/42 facilitated signaling pathways. In addition, LSC treatment increased insulin-sensitizer adiponectin expression and reduced insulin resistance-related proteins such as resistin and aP2. Cells treated with LSC for 12 h showed enhanced lipolysis by increasing phosphorylation of AMPK-α at Thr172 and inhibiting lipogenesis-associated enzymes FAS and ACC. Overall, the results of this study suggest that LSC has potential as very effective multifunctional probiotic bacteria with anti-obesity activity. In order to fully understand the potential role of LSC in the gastrointestinal tract, more intensive studies are needed to confirm the potential role of LSC in the inhibition of obesity and its associated diseases/disorders in animal or human model experiments in the future.

## Figures and Tables

**Figure 1 ijms-24-07367-f001:**
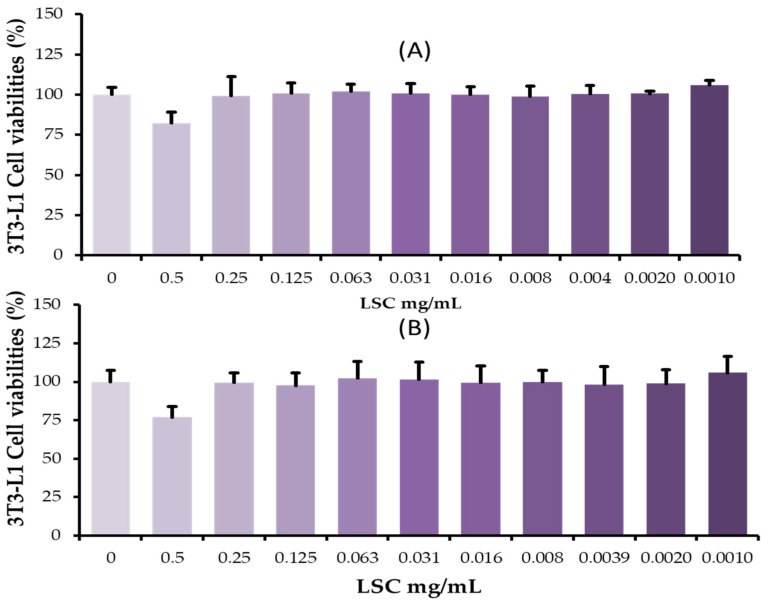
Effects of *L. citreum* cell-free supernatants on 3T3-L1 cells. Cells were incubated with different concentrations of LSC under normal cell culture conditions. After 24 h and 48 h, EZcytox reagent was used to determine cell viability. (**A**) Percentage of viable cells in experimental groups at 24 h; (**B**) Percentage of viable cells at 48 h. Data are expressed as mean (SD) of five replicates.

**Figure 2 ijms-24-07367-f002:**
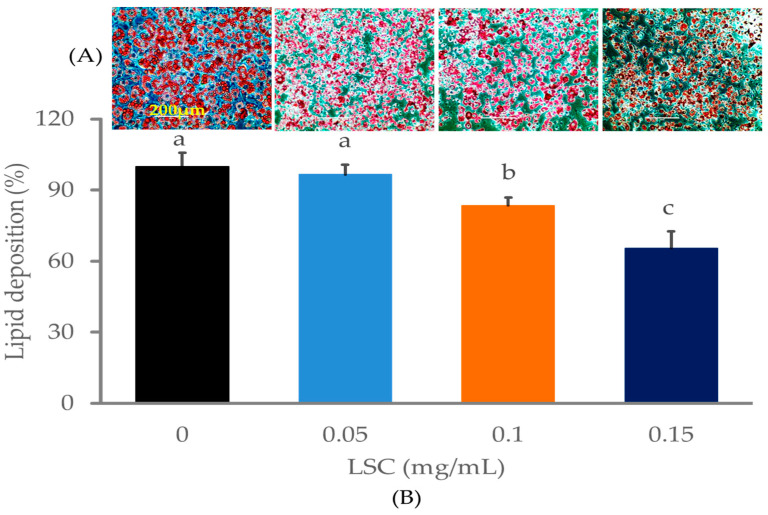
Impact of LSC on fat deposition in adipocytes on day 10. Cells were seeded in a 6-well culture plate at the density of 3 × 10^4^/well and incubated at 37 °C with 5% CO_2_. Differentiation was induced by differentiation cocktails (insulin, DEX and IBMX) for 48 h. The cells were then switched to insulin medium for another 48 h. LSC at different concentrations was used to treat cells when differentiation was initiated. ORO staining was performed to stain lipid depositions in differentiated cells. (**A**) Oil red O-stained cells (200 μm) were then observed using an Evos microscope (20×/20× magnifications). (**B**) Percentage of fat depositions in experimental adipocytes. Data are presented as mean ± standard deviation (*n* = 5). Different alphabets within the figure indicate significant differences at *p* < 0.05 level.

**Figure 3 ijms-24-07367-f003:**
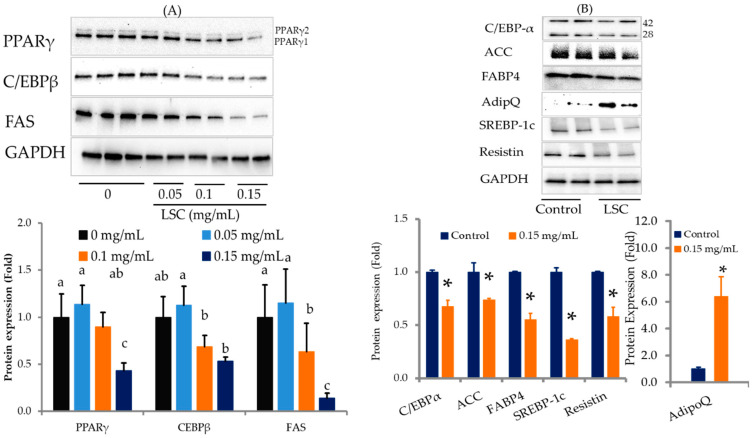
Effects of LSC on adipogenesis and lipogenesis in 3T3-L1 cells on day 10. Proteins were extracted with protease and phosphatase inhibitors and quantified using BCA on day 10. Proteins were separated by SDS-PAGE. PPAR-γ, C/EBP-β, C/EBP-α, and SREBP-1c; lipogenic proteins such as FAS, and ACC, insulin sensitizer adiponectin (AdipoQ) in insulin resistance-inducing proteins such as FABP4 (aP2) and resistin were detected with specific antibodies using Western blot. Protein intensity was quantified with ImageJ software, 1.49 versions (32-bit). (**A**) Preliminary screening of key adipogenic and lipogenic proteins expression changes in experimental cells. (**B**) Impact of selected concentration of LSC on other proteins involved in adipogenesis and lipogenesis. Results are expressed as mean ± standard deviation of three replicates (*n* = 3). * *p* < 0.05. Different alphabets in the figure indicate significant differences at *p* < 0.05.

**Figure 4 ijms-24-07367-f004:**
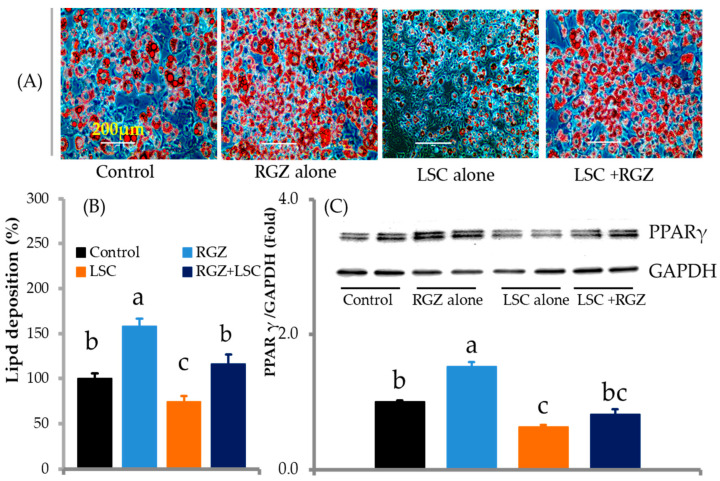
LSC treatment aborts rosiglitazone (RGZ)-induced differentiation and fat deposition. Cells were incubated with LSC (0.15 mg/mL) or RGZ (1 µM) or RGZ plus LSC during differentiation. RGZ treatment alone increased differentiation and lipid synthesis in adipocytes, whereas LSC treatment alone reduced fat accumulation. Furthermore, LSC inhibits RGZ-induced lipid accumulation. (**A**) Microscopic views (20×/20× magnifications) of ORO-stained fat deposition in adipocytes. (**B**) Percentage of lipid accumulation in experimental cells (*n* = 5). (**C**) PPAR-γ protein expression in experimental cells on day 10 (*n* = 3). Data are presented as mean ± standard deviation. Different alphabets in the figure indicate significant differences (*p* < 0.05).

**Figure 5 ijms-24-07367-f005:**
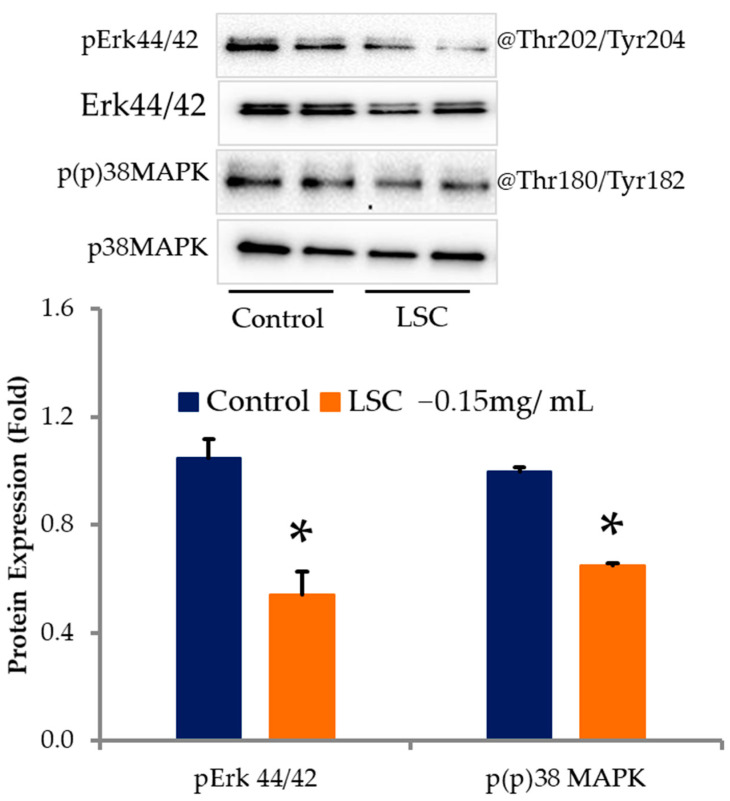
LSC modulates phosphorylating levels of pp38MAPK and pERK 44/42 in adipocytes. On day 10, proteins were extracted with RIPA buffer containing protease and phosphatase inhibitors and quantified with BCA. SDS-PAGE was used to separate proteins. Erk 44/42 and p38MAPK phosphorylation levels were determined with specific antibodies. ImageJ was used to quantify protein intensity. Results are expressed as mean ± standard deviation. * *p* < 0.05 (*n* = 3).

**Figure 6 ijms-24-07367-f006:**
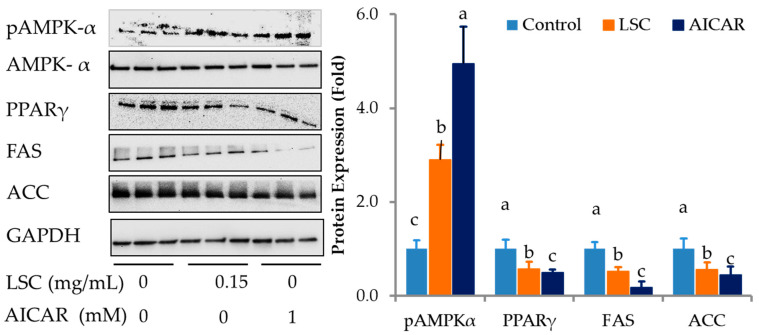
Effects of LSC on expression levels of lipolysis and lipogenesis related proteins in adipocytes. Differentiated adipocytes for eight days were treated with LSC or AICAR, an AMPKα agonist, for 12 h. Phosphorylation level of AMPK-α, a key adipocyte differentiation transcriptional factor PPAR-γ, and lipogenic enzymes such as FAS and ACC were determined using Western blot. Protein intensity was determined using ImageJ software. Data are expressed as mean ± standard deviation of three replicates. Different alphabets in the figure indicate significant differences at *p* < 0.05.

## Data Availability

Experimental data can be obtained from the corresponding author upon reasonable request.

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
