# Peer review of "Leuconostoc Citreum Inhibits Adipogenesis and Lipogenesis by Inhibiting p38 MAPK/Erk 44/42 and Stimulating AMPKα Signaling Pathways"

_ijms, 2023, doi:10.3390/ijms24087367_

Round 1

Reviewer 1 Report

1. Figure 1: statistical analysis and error bars should be added.

2. Why did the author select 0.05, 0.1, 0.15 mg/mL of LSC for the experiments, not the concentrations which were examined in cell viability assays?

3. Figure 3A is not fully visible. Please adjust it.

4. Figure 4: please re-organize the order of Figure 4A, B, C. Figure 4A should be on the upper side.

5. Figure 5A: pMAPK38 and MAPK38 should be pp38MAPK and p38MAPK.

Figure 5B: Please replace the pAMPK-α and PPAR-γ bands with better blots.

Please separate the graph by each protein. The meaning of letters in this graph seems confusing.

6. There are many incorrect information in the manuscript:

Line 66: DMEM should be MRS.

Line 114, 124: C/CEB should be C/EBP.

Line 132: PPAR-γ2 should be PPAR-γ.

Line 161: pp38MAPK and pERK 44/42 should be p38MAPK and ERK 44/42.

Line 246-252: ERK belongs to MAPK family, so "MAPKs and ERK1/2" is not correct.

7. Section 2.5: LSC inhibits differentiation and fat deposition through p38MAPK and Erk1/2 signaling. Please provide more evidence to prove this statement as well as discuss this more detailed in the Discussion part.

Author Response

We thank the reviewer for providing useful comments about our research paper, which will greatly improve the quality of the manuscript. We ask apology for typographical mistakes and grammatical errors. In response to the reviewers' suggestions, we have read through the entire manuscript and modified it accordingly. Please note that changes have been made in red across the manuscript.

  1. Figure 1: statistical analysis and error bars should be added.

Thank you for your kind information. Now, the standard error for figure 1 has        been provided.

2.  Why did the author select 0.05, 0.1, 0.15 mg/mL of LSC for the experiments, not the concentrations which were examined in cell viability assays?

Less than 0.5mg/mL of cell free metabolites did not influence the cell viability. Therefore, we have selected the lower concentration 0.05 mg/mL and higher concentration 0.15 mg/mL for preliminary study.

3. Figure 3A is not fully visible. Please adjust it.

Yes, It has now revised as fully visible mode.

4. Figure 4: please re-organize the order of Figure 4A, B, C. Figure 4A should be on the upper side.

Yes, we strongly agreed with the reviewer comment and revised the same in the manuscript.

5. Figure 5A: pMAPK38 and MAPK38 should be pp38MAPK and p38MAPK.

Yes, we strongly agreed with the reviewer comment and revised the same in the manuscript.

6. Figure 5B: Please replace the pAMPK-α and PPAR-γ bands with better blots.

Yes, we understand the reviewer thoughts. But, current situation we cannot replace it  due to out of office.

  1. Please separate the graph by each protein. The meaning of letters in this graph seems confusing.

Yes each figure has been separated well to understand easily.

8. There are many incorrect information in the manuscript:

We would like to ask apology for these inconveniences. Now the language of the manuscript has been edited by language editing service.

9. Line 66: DMEM should be MRS.

In this present study, LSC was grown in 10%FBS containing DMEM 30-2002 without antibiotics for production of secondary metabolites productions instead of MRS medium.

  1. Line 114, 124: C/CEB should be C/EBP. Line 132: PPAR-γ2 should be PPAR-γ.

Thanks, we have revised protein abbreviation correctly in throughout the manuscript.

  1.  

Line 161: pp38MAPK and pERK 44/42 should be p38MAPK and ERK 44/42.Line 246-252: ERK belongs to MAPK family, so "MAPKs and ERK1/2" is not correct.

Thanks, we have revised protein abbreviation correctly in throughout the manuscript.

  1. Section 2.5: LSC inhibits differentiation and fat deposition through p38MAPK and Erk1/2 signaling. Please provide more evidence to prove this statement as well as discuss this more detailed in the Discussion part.

 Adipocyte differentiation is directed by p38MAPK and ERK1/2 [57]. By inhibiting p38MAPK and ERK1/2 phosphorylation, specific inhibitors have been shown to decrease adipocyte differentiation and lipid accumulation [58–60], suggesting that both p38MAPK and ERK1/2 initiations are crucial to lipogenesis and adipogenesis. Cells treated with LSC showed lower levels of phosphorylation of p38MAPK at Thr180/Tyr182 as well as Erk1/2 at Thr202/Tyr204 than control cells, which inhibits key transcriptional factors PPAR, C/EBP-α and C/EBP-β as well as lipogenic associated enzymes FAS and ACC [24].

Reviewer 2 Report

Specific comments:

1.       Line 20-21:

Please mention what type of adipocytes you are referring to.

2.       Line 58:  “modulating appetite. [13,15–17].”

Please, delete point.

3.       Line 130: “cells. (Figure 5A).”

Please, change comma to point, after parenthesis.

4.       Line 132

 What is PPAR-γ2? please correct and delete number 2, if it applicable.

5.       Line 159: “cells. (Figure 5A).”

Please delete point after “cells”

6.       Line 162

Please improve the wording, redundant words (extracted in extraction)

7.       Linea 174

Please, correct to “an AMPKα agonist”

8.       Line 184: “health. [31].”

Please, delete point.

9.       Line 186:  “obesity. [32].”

Please, delete point.

10.   Line 301-310

Please use the acronym of Oil Red O

11.   Line 333: MPAK

Please, put the correct form to MPAK

12.   Figure 3A

The last GAPDH band is incomplete. Please show the full figure.

13.   Figure 4c

Please unify RGZ nomenclature.

14.   Figure 4c

Please decrease the size of the legend on the Y axis

15.   General text

Please remove capital letter in Resistin

16.   Figures

Please indicate the measurement of the bar in the photograph of the cells

17.   General text

Please unify the space between the parentheses of the reference throughout the text

18.   General text

Please, unify C/EBP nomenclature.

19.   General text

Please, unify PPARγ nomenclature both in text and in figures.

20.   General text

Please, unify AMPKα nomenclature both in text and in figures.

21.   Please explain both in results and in discussion the function of adipoQ

22.   It is suggested in conclusions, discuss further the use of adipogenesis inhibitors and their impact on weight loss either in animal models or in human studies.

Author Response

We thank the reviewer for providing useful comments about our research paper, which will greatly improve the quality of the manuscript. We ask apology for typographical mistakes and grammatical errors. In response to the reviewers' suggestions, we have read through the entire manuscript and modified it accordingly. Please note that changes have been made in red across the manuscript.

  1. Line 20-21:Please mention what type of adipocytes you are referring to.

Therefore, the objective of this study was to determine effects of cell-free metabolites of L. citreum (LSC) on adipogenesis, lipogenesis, and lipolysis in 3T3-L1 adipocytes

  1. Line 58:  “modulating appetite. [13,15–17].Please, delete point.

Yes deleted

  1. Line 130: “cells. (Figure 5A).”Please, change comma to point, after parenthesis.

Yes, changed comma to point after parenthesis

  1. Line 132  What is PPAR-γ2? please correct and delete number 2, if it applicable.

Yes, deleted 2 from PPAR gamma

  1. Line 159: “cells. (Figure 5A).” Please delete point after “cells”

Yes, point has been deleted.

  1. Line 162 Please improve the wording, redundant words (extracted in extraction)

Thanks, We revised these lines as Figure 5. A. LSC modulates phosphorylating levels of pp38MAPK and pERK 44/42 in adipocytes. On day 10, proteins were extracted with RIPA buffer containing protease and phosphatase inhibitors and quantified with BCA. SDS-PAGE was used to separate proteins. Erk 44/42 and p38MAPK phosphorylation levels were determined with specific antibodies. ImageJ was used to quantify protein intensity. Results are expressed as mean ± standard deviation. * p < 0.05 (n = 3).

  1. Linea 174 Please, correct to “an AMPKα agonist”

Differentiated adipocytes treated with LSC at 0.15 mg/mL or AICAR (5-Aminoimidazole-4-carboxamide ribonucleotide) at 1 mM, an AMPK-α agonist, for 12 h increased phosphorylation of AMPK-α closely associated with lipolysis but decreased PPAR-γ, FAS, and ACC expression levels closely associated with adipocyte differentiation and fatty acid synthesis (Figure 5B). Results obtained for LSC were significantly comparable with AICAR compared to control cells.

Differentiated adipocytes for eight days were treated with LSC or AICAR, an AMPKα agonist, for 12 h

  1. Line 184: “health. [31].”Please, delete point.

Yes, deleted.

  1. Line 186:  “obesity. [32].”Please, delete point.

Yes, deleted.

  1. Line 301-310

Please use the acronym of Oil Red O

Yes, we have provided acronym for Oil Red O as Detection and determination of fat deposition by Oil Red O (ORO) staining

  1. Line 333: MPAK Please, put the correct form to MPAK

Now these lines have been revised as Cell-free metabolites from Leuconostoc citreum (LSC) could inhibit adipogenesis and lipogenesis by downregulating the expression of main differentiation transcriptional factors PPAR-γ , C/EBP α, and C/EBP β and vital lipogenic enzymes FAS and ACC through p38 MAPK and Erk 44/42 facilitated signaling pathways

  1. Figure 3A The last GAPDH band is incomplete. Please show the full figure.

Thanks. We have provided clear image for figure A

  1. Figure 4c Please unify RGZ nomenclature.

The RGZ nomenclature has been revised uniformely.

  1. Figure 4c Please decrease the size of the legend on the Y axis

Yes. We agreed with the reviewer comment and revised the same.

  1. General text Please remove capital letter in Resistin

The resistin name has been revised as per the reviewer suggestion.

  1. General text Please unify the space between the parentheses of the reference throughout the text

Thank you for your suggestion. We have provided space between the parentheses of the reference

  1. General text Please, unify C/EBP nomenclature.

unified all C/EBP nomenclature throughout manuscript

  1. General text Please, unify PPARγ nomenclature both in text and in figures.

unified all PPARγ nomenclature throughout manuscript

  1. General text Please, unify AMPKα nomenclature both in text and in figures.

unified all AMPKα nomenclature throughout manuscript

  1. Please explain both in results and in discussion the function of adipoQ

 Yes . we have interpreted adipoQ  data as Adipokines such as adiponectin (AdipoQ), resistin, and leptin are physiological High adiponectin levels in the circulation are associated with lower incidence of diabetes [53,54].

  1. It is suggested in conclusions, discuss further the use of adipogenesis inhibitors and their impact on weight loss either in animal models or in human stud

In order to fully understand the potential role of LSC in the gastrointestinal tract, more intensive studies are needed to confirm the potential role of LSC in inhibition of obesity and its associated diseases/disorders in animal or human model experiments in the future.

Reviewer 3 Report

Cell-free metabolites from Leuconostoc citrum (LSC) inhibited adipogenesis and lipogenesis via downregulating the expression of main differentiation transcriptional factors PPAR γ, C/EBP α, and C/EBP β and the vital lipogenic enzymes FAS, and ACC by p38 MPAK and Erk 44/42 facilitated signaling pathways.  Overall, the study suggested that LSC has the potential to become the most effective multifunctional probiotic bacteria with anti-obesity activity.

As a weakness of the article, the authors themselves define it: In order to fully understand the potential role of LSC in the gastrointestinal tract, more intensive studies are needed.

The conclusions are consistent with the evidence presented. Definitely, the authors address the main question posed. The bibliographical references are current and cover a wide range of articles that support the research developed. 

The Tables and Figures contain the correct scientific information with the necessary statistical parameters to give them credibility.

Author Response

We thank the reviewer for providing useful comments about our research paper, which will greatly improve the quality of the manuscript. We ask apology for typographical mistakes and grammatical errors. In response to the reviewers' suggestions, we have read through the entire manuscript and modified it accordingly. Please note that changes have been made in red across the manuscript.

Cell-free metabolites from Leuconostoc citreum (LSC) inhibited adipogenesis and lipogenesis via downregulating the expression of main differentiation transcriptional factors PPAR γ, C/EBP α, and C/EBP β and the vital lipogenic enzymes FAS, and ACC by p38 MPAK and Erk 44/42 facilitated signaling pathways.  Overall, the study suggested that LSC has the potential to become the most effective multifunctional probiotic bacteria with anti-obesity activity.

As a weakness of the article, the authors themselves define it: In order to fully understand the potential role of LSC in the gastrointestinal tract, more intensive studies are needed.

In this study we determined the survival ability of LSC in artificial simulated gastric juices in PBS. In order to fully understand the potential role of LSC in the gastrointestinal tract, more intensive studies are needed to confirm the potential role of LSC in inhibition of obesity and its associated diseases/disorders in animal or human model experiments in the future.

The conclusions are consistent with the evidence presented. Definitely, the authors address the main question posed. The bibliographical references are current and cover a wide range of articles that support the research developed. 

Thank you for your positive comments

The Tables and Figures contain the correct scientific information with the necessary statistical parameters to give them credibility.

Thank you for your positive comments

Reviewer 4 Report

Introduction.

The second paragraph needs significant restructuring. Please add a summary of the hypothesis at the end. Also, please describe clearly the objectives of the study.

Procedures

These are ok, I have no comments for potential changes.

Results

Figure 1. I suggest to colourise the bars with progressively darkening shades to increase impact of the graph.

Same for figure 2.

In the figures, please try to maintain same colours for same variable throughout the manuscript, in order to have a uniformity in the presentation of results.

Discussion

1. The discussion can be usefully divided into two sections, for easier flow of the reading.

2. Some recent relevant references, which are missing, can help the authors to better explain their findings.

Conclusion

Please do not present new ideas in this section (these should be entered in the discussion). Only present the main message from the findings of the study.

Author Response

We thank the reviewer for providing useful comments about our research paper, which will greatly improve the quality of the manuscript. We ask apology for typographical mistakes and grammatical errors. In response to the reviewers' suggestions, we have read through the entire manuscript and modified it accordingly. Please note that changes have been made in red across the manuscript.

Introduction.

  1. The second paragraph needs significant restructuring. Please add a summary of the hypothesis at the end. Also, please describe clearly the objectives of the study.

Therefore, in the present study, we used a novel strain of Leuconostoc citreum to produce secondary metabolites (LSC) in DMEM (Dulbecco's Modified Eagle Medium) and lyophilized. Effects of cell-free supernatant from L. citreum (LSC) on adipocyte differentiation and fat deposition were investigated. Additionally, molecular mechanisms involved in inhibition of lipogenesis and lipolysis by LSC were examined

Procedures

  1. These are ok, I have no comments for potential changes.

Thank you for your positive comments

 Results

  1. Figure 1. I suggest to colourise the bars with progressively darkening shades to increase impact of the graph.

Thank you for your valuable suggestions. We have modified the figure 1 as per reviewer suggestions

Same for figure 2.

  1. In the figures, please try to maintain same colours for same variable throughout the manuscript, in order to have a uniformity in the presentation of results.

 Thank you for your valuable suggestions. We have modified the figures as per reviewer suggestions

Discussion

  1. The discussion can be usefully divided into two sections, for easier flow of the reading.

Yes, strongly agreed with reviewer suggestion. First we discussed general parameters which are related to adipogenesis and then molecular markers that are associated with obesity were interpreted.

  1. Some recent relevant references, which are missing, can help the authors to better explain their findings

We have provided several references which are related to current study (More than 60 references)

 Conclusion

  1. Please do not present new ideas in this section (these should be entered in the discussion). Only present the main message from the findings of the study.

Cell-free metabolites from Leuconostoc citreum (LSC) could inhibit adipogenesis and lipogenesis by downregulating the expression of main differentiation transcriptional factors PPAR-γ , C/EBP α, and C/EBP β and vital lipogenic enzymes FAS and ACC through p38 MAPK and Erk 44/42 facilitated signaling pathways. In addition, LSC treatment increased insulin-sensitizer adiponectin expression and reduced insulin resistance related proteins such as resistin and aP2. Cells treated with LSC for 12 h showed enhanced lipolysis by increasing phosphorylation of AMPK-α at Thr172 and inhibiting lipogenesis-associated enzymes FAS and ACC. Overall, results of this study suggest that LSC has potential as very effective multifunctional probiotic bacteria with anti-obesity activity. In order to fully understand the potential role of LSC in the gastrointestinal tract, more intensive studies are needed to confirm the potential role of LSC in inhibition of obesity and its associated diseases/disorders in animal or human model experiments in the future.

Based on other reviewers, we have revised this conclusion.

Reviewer 5 Report

The manuscript reports important data about some exploitable benefits of using Leuconostoc citreum cell-free supernatant for reducing or inhibition of adipoensis and lipogenesis. Somewhere in the introduction, the authors claim that to their best knowledge, this would be the first report demonstrating that L. c. affects adipogenesis and lipogenesis. However, in a short literature search, I found for instance this article:

Kim et al., 2008, Lipid profile lowering effect of SoyproTM fermented with lactic acid bacteria isolated from Kimchi in high-fat diet-induced obese rats. BioFactors 33, 49-60.

In that publication, a soy milk was fermented with lactic acid bacteria, among which L. c., and the effect of this product on plasma lipid levels and weight gain in rats and effect on adipocyte differentiation in pre-adipocyte 3T3-L1 cell lines.-  Most likely, there is more literature existing on reporting similar effects in other fermented products. I would suggest that in the revised document, the authors cite this one reference and underline the difference of their study (e.g. cell-free supernatant of L.c. alone compared to a milk-based product that was produced using a combination of different lactic acid bacteria.

As editorial remark, there are a few singular/plural confusions, as well as using upper cases in some instances where lower cases would be appropriate.

Specific comments:

Abstract:

Line 25, resistin … In contract, compared to control cells, adiponectin, … treated with LSC.

Introduction

Line 35, … 10.7 % obese in their populations.

Lien 39, According to Asper, .. What is Asper?

Line 41, Jung et al. 2020 (no first names in citations, please).

Line 47, … globally in context with obesity and related diseases.

Lines 65-66, the sentence seems to be fragmentary.

Line 66, please explain DMEM.

Line 72, … report showing that …

Results

Figure 1, why are the standard deviations not shown as error bars?

Line 82, citreum

Line 91, insulin, dexamethasone

Line 95, strongly inhibited; well, ca. 30 % reduced compared to the control (Figure 2)

Line 110, downregulates

Line 114, CEB-a, should this be EBP-a?

Line 117, FAS and aP2 not shown in Figure 3B.

Line 119, in adipocytes compared to …

Figure 3, A and B (WBs): why are there 3 controls left side and 2 controls right side? A, 0.05 mg/mL was the only concentration tested, why this one? What was the concentration in 3B?

Line 124, again, CEB-a or b, not shown in Figure 3, should this be EBP-a and b?

Figure 3 upper left side, what is CE?

Line 132, RGZ

Line 136, RGZ treatment alone

Figure 4C, why two WB samples per molecule tested? Rosi should be RGZ. What shall the bands for GAPDH indicate?

Figure 5A, why again two samples for control and for LSC? Why are the results described and shown in the column chart do not fit the WB results (the bands in the WB seem to be of almost identical intensity)?

Line 158, pp38 MAPK or p38 MAPK? pERK 44/42 or Erk ½?

Line 167, AICAR?

Why Figure 5A and 5B and not 5 and 6?

Figure 5B right part, column chart, x-axis, shouldn’t it be p(p)MAPK and not AMPK?

Discussion

Line 211, LSC and RGZ seem to counteract, so there’s almost no difference between control and LSC+RGZ as shown in Figure 4C.

Line 211, whereas

Line 238, in contrast,

Line 262, activates

Line 263, PPAR and ACC effect, again, the WB results look different (comparable to same intensities) compared to the column chart on the right side of the Figure.

Line 270, was comparable

Materials and Methods

Line 279, followed by lyophilisation. How long was sample lyophilized? 50 mTorr (about 0.07 mbar)

Line 287, Then microliters of

Line 318, semi-dry

Conclusions

Line 335, enhanced lipolysis? Reduced it should be.

Line 338, a very effective

References

Please use journal abbreviations

Author Response

We thank the reviewer for providing useful comments about our research paper, which will greatly improve the quality of the manuscript. We ask apology for typographical mistakes and grammatical errors. In response to the reviewers' suggestions, we have read through the entire manuscript and modified it accordingly. Please note that changes have been made in red across the manuscript.

  1. The manuscript reports important data about some exploitable benefits of using Leuconostoc citreum cell-free supernatant for reducing or inhibition of adipoensis and lipogenesis. Somewhere in the introduction, the authors claim that to their best knowledge, this would be the first report demonstrating that L. c. affects adipogenesis and lipogenesis. However, in a short literature search, I found for instance this article:

Kim et al., 2008, Lipid profile lowering effect of SoyproTM fermented with lactic acid bacteria isolated from Kimchi in high-fat diet-induced obese rats. BioFactors 33, 49-60.

In that publication, a soy milk was fermented with lactic acid bacteria, among which L. c., and the effect of this product on plasma lipid levels and weight gain in rats and effect on adipocyte differentiation in pre-adipocyte 3T3-L1 cell lines.-  Most likely, there is more literature existing on reporting similar effects in other fermented products. I would suggest that in the revised document, the authors cite this one reference and underline the difference of their study (e.g. cell-free supernatant of L.c. alone compared to a milk-based product that was produced using a combination of different lactic acid bacteria.

Really thank you for your valuable information’s. We did not use any fermented products production in the presence of L. citrum for antiobesity activity. We used secondary metabolites produced by LC in DMEM for the current study. Any have we strongly agreed with reviewer comment and revised it correctly.Therefore, in the present study, we used a novel strain of Leuconostoc citrum to produce secondary metabolites (LSC) in DMEM and lyophilized. Effects of cell-free supernatant from L. citrum (LSC) on adipocyte differentiation and fat deposition were investigated. Additionally, molecular mechanisms involved in inhibition of lipogenesis and lipolysis by LSC were examined.

  1. As editorial remark, there are a few singular/plural confusions, as well as using upper cases in some instances where lower cases would be appropriate.

The language of the manuscript has been edited by language experts by harrisco pvt and evidence also attached in cover letter.

Specific comments:

Abstract:

  1. Line 25, resistin … In contract, compared to control cells, adiponectin, … treated with LSC.

In contrast, compared to control cells, adiponectin, an insulin sensitizer, was elevated in adipocytes treated with LSC

Introduction

  1. Line 35, … 10.7 % obese in their populations.

Globally, obesity is becoming an epidemic issue, with an increasing occurrence rate. Obesity rate is 30.4% in the United States, 12.8% in Europe, and 10.7% in their populations in China

  1. Lien 39, According to Asper, .. What is Asper?

Asper Organization for Economic Co-operation & Development, more than 4% and almost 30% of the adult population in Korea are obese and overweight, respectively

  1. Line 47, … globally in context with obesity and related diseases.

More than 2.8 million adults have been dying each year globally in context with obesity and related diseases

  1. Lines 65-66, the sentence seems to be fragmentary.

Now these sentences have been revised as Several studies have claimed that Leuconostoc spp. can mediate food supplements and reduce obesity and metabolic diseases associated with obesity. Therefore, in the present study, we used a novel strain of Leuconostoc citrum to produce secondary metabolites (LSC) in DMEM and lyophilized. Effects of cell-free supernatant from L. citrum (LSC) on adipocyte differentiation and fat deposition were investigated. Additionally, molecular mechanisms involved in inhibition of lipogenesis and lipolysis by LSC were examined.

  1. Line 66, please explain DMEM.

Yes, we have explained as Therefore, in the present study, we used a novel strain of Leuconostoc citrum to produce secondary metabolites (LSC) in DMEM (Dulbecco's Modified Eagle Medium)

Results

  1. Figure 1, why are the standard deviations not shown as error bars?

Now the standard error bars have been provided for figure 1.

  1. Line 82, citreum

Thanks. We have revised the name of the probiotic in whole manuscript.

  1. Line 91, insulin, dexamethasone

Yes revised as insulin, IBMX, and dexamethasone

  1. Line 95, strongly inhibited; well, ca. 30 % reduced compared to the control (Figure 2)

These lines have been revised

  1. Line 110, downregulates

Yes, revised as downregulates

  1. Line 114, CEB-a, should this be EBP-a?

Sorry. It is typographical error. Now it has been revised correctly

  1. Line 117, FAS and aP2 not shown in Figure 3B.

aP2 another name is FABP4. Now these lines have been revised as LSC treatment also inhibited the expression of key lipogenesis associated proteins such as FAS, ACC, and aP2. Other proteins related to insulin resistance and insulin sensitivity such as Resistin and adiponectin levels were also determined. Results showed that LSC treatment decreased Resistin but increased adiponectin expression in adipocytes than control (Figure 3A-B).

  1. Line 119, in adipocytes compared to …

Figure 3, A and B (WBs): why are there 3 controls left side and 2 controls right side? A, 0.05 mg/mL was the only concentration tested, why this one? What was the concentration in 3B?

First, we determined key adipocyte differentiation markers PPAR and CEBP beta and key lipogenic enzyme fatty acid synthase at multiple concentrations to find a suitable dose for further analysis. Therefore, we used the additional controls.

  1. Line 124, again, CEB-a or b, not shown in Figure 3, should this be EBP-a and b?

. Western blot was used to determine molecular mechanisms involved in LSC treatment-induced fat reduction in adipocytes. Key transcriptional factors such as PPAR-γ, C/EBP-α/β, and SREBP-1c associated with differentiation were downregulated in adipocytes treated with LSC on day 10 (Figure 3A). LSC treatment also inhibited the expression of key lipogenesis associated proteins such as FAS, ACC, and aP2. Other proteins related to insulin resistance and insulin sensitivity such as resistin and adiponectin levels were also determined. Results showed that LSC treatment decreased resistin but increased adiponectin expression in adipocytes compared to control (Figure 3A-B).

  1. Line 132, RGZLine 136, RGZ treatment alone

Yes Revised as RGZ treatment alone rapidly increased fat deposition in adipocytes by upregulating PPAR-γ expression compared to control.

  1. Figure 4C, why two WB samples per molecule tested? Rosi should be RGZ. What shall the bands for GAPDH indicate?

Here we used two independent replicate samples were used to confirm the expression of master protein PPAR expressions. GAPDH  is a house keeping gene can be used to calculate fold of target protein expression after normalization with GAPDH. 

  1. Figure 5A, why again two samples for control and for LSC? Why are the results described and shown in the column chart do not fit the WB results (the bands in the WB seem to be of almost identical intensity)?

There is no identical intensity between groups. After normalized with house keeping gene GAPDH, this result showed some significant changes in targeted protein expression. Here we used two independent replicate samples were used to confirm the expression of master protein PPAR expressions. GAPDH  is a house keeping gene can be used to calculate fold of target protein expression after normalization with GAPDH. 

  1. Line 158, pp38 MAPK or p38 MAPK? pERK 44/42 or Erk ½?

pp38 MAPK. It indicates phosphorylated status of p38;

p38 MAPK it indicates non-phosphorylated status.

Erk 44/42 or Erk ½  both are same.

  1. Line 167, AICAR?

Differentiated adipocytes treated with LSC at 0.15 mg/mL or AICAR (5-Aminoimidazole-4-carboxamide ribonucleotide) at 1 mM, an AMPK-α agonist, for 12 h increased phosphorylation of AMPK-α closely associated with lipolysis but decreased PPAR-γ, FAS, and ACC expression levels closely associated with adipocyte differentiation and fatty acid synthesis (Figure 5B). Results obtained for LSC were significantly comparable with AICAR compared to control cells

  1. Why Figure 5A and 5B and not 5 and 6?

Both figures are related to molecular signalling’s

  1. Figure 5B right part, column chart, x-axis, shouldn’t it be p(p)MAPK and not AMPK?

Currently reported is correct .phosphorylated AMPK (pAMPK) and total AMPK ( AMPK).

Discussion

  1. Line 211, LSC and RGZ seem to counteract, so there’s almost no difference between control and LSC+RGZ as shown in Figure 4C.

Yes definitely, there is no significant difference between control and LSC+RGZ treatments. But, compared RGZ treatment, LSC+RGZ treatment had significant reduction at 0.05level.

  1. Line 211, whereas Line 238, in contrast,Line 262, activates

Revised all

  1. Line 263, PPAR and ACC effect, again, the WB results look different (comparable to same intensities) compared to the column chart on the right side of the Figure.

Yes definitely, there is no significant difference between control and LSC+RGZ treatments. But, compared RGZ treatment, LSC+RGZ treatment had significant reduction at 0.05level.

  1. Line 270, was comparable

Yes revised as by increasing its phosphorylating level. Result after LSC treatment was comparable to those after AICAR treatment

Materials and Methods

  1. Line 279, followed by lyophilisation. How long was sample lyophilized? 50 mTorr (about 0.07 mbar)

Now it has revised as filtrating with filter membranes having different pour sizes, and then lyophilizing at -40 °C under less than 50 m Torr pressure for 72h

  1. Line 287, Then microliters of Line 318, semi-dry

Now revised as Afterwards, proteins were transferred to polyvinylidene difluoride membranes (PVDF) using a semiwet transfer method

Conclusions

  1. Line 335, enhanced lipolysis? Reduced it should be.

Thank you for your kind information and revised as Cells treated with LSC for 12 h showed enhanced lipolysis by increasing phosphorylation of AMPK-α at Thr172 and inhibiting lipogenesis-associated enzymes FAS and ACC. It is correct hypothesis. In general, increases of lipolysis means breakdown of lipid has been increased. So, it is an essential to stimulate the lipolysis to reduce body fat content.

  1. Line 338, a very effective

Yes, revised as very effective

References Please use journal abbreviations

We have used endnote software with MDPI endnote style for reference preparations.

Round 2

Reviewer 1 Report

The authors resolved all mentioned issues.

The revised version is appropriate for publication.

Author Response

I would like to thank you for your positive comments regarding our manuscript that we submitted

Reviewer 4 Report

The manuscript has been revised correctly.

Before final acceptance, I suggest to add a new paragraph in the discussion, presenting the clinical implications of this work.

Author Response

We thank the reviewers for their critical and judicious evaluation of our manuscript, and for providing constructive suggestions for improving its quality. All reviewers' comments have been carefully considered and the manuscript has been thoroughly revised. I have responded to the reviewer's comments point by point. Red fonts were used for all changes in manuscript.

1.Before final acceptance, I suggest to add a new paragraph in the discussion, presenting the clinical implications of this work.

Thank you for your valuable suggestion and revised it same as

Several researchers have actively involved in production of fermented products in the presence of Leuconostoc spp with significant biological potential. As examples, the soymilk fermented with L. kimchi, L. citreum and L. plantarum significantly reduced fat deposition in 3T3-L1 adipocytes by inhibiting key transcription factors C/EBP-α and PPAR-γ. Furthermore, Soypro treatment reduced low density lipoprotein cholesterol (LDL) levels without affecting body weight in obese rats (Kim et al., 2008). Another study reported that pear extract and robusta fermented with L. mesenteroids significantly reduced body weight and adipose tissue mass. Also reduced the size of lipids in liver in obese rats compared to control rats (Choi et al., 2017; Chu & Kim, 2018). Supplementation with L. mesenteroids reduced blood urea nitrogen, glucose, and triglycerides levels in obese mice serum, as well as fatty liver development  and liver steatosis in comparison with controls (Castro-Rodríguez et al., 2020; Lee et al., 2018). Thus, we investigated effects of LSC on fat deposition and differentiation in 3T3-L1 adipocytes. But a significant difference exists between this study and previous studies on Leuconostoc species.

Reviewer 5 Report

Several change requests were not addressed (text remained unchanged), for instance line 42 (the first name of the leading author was not deleted), line 70-71 (“and lyophilised.” is meaningless), line 318, line 335 and still using full journal names instead of abbreviations. Moreover, none of my questions was answered is usually would be the approach in other journals (the authors address item-by-item the reviewer’s comments and explain what they changed (and/or did not change and why). Especially my queries regarding Figures 3, 4, 5A and B, exact abbreviations (e.g. line 263) remained un-answered. Also, my initial remark to cite the Kim publication and refer to the differences in the present work was half neglected (the claim that “this is the first report that L.c. affects adipogenesis and lipogenesis” has now been dropped).

I recommend a third review round with proper addressing the open queries, as otherwise I would recommend to reject the manuscript. 

Author Response

At the outset we thank the reviewers for their critical and judicious evaluation of our manuscript, and providing constructive suggestions for improving the quality and presentation of the manuscript. We have carefully considered the comments of the reviewers and revised the manuscript thoroughly taking all the points into consideration. Point wise response to the reviewer's comments is given below. All changes in manuscript were made with red color fonts.

Responses to reviewer comments

Several change requests were not addressed (text remained unchanged), for instance line 42 (the first name of the leading author was not deleted), line 70-71 (“and lyophilised.” is meaningless), line 318, line 335 and still using full journal names instead of abbreviations. Moreover, none of my questions was answered is usually would be the approach in other journals (the authors address item-by-item the reviewer’s comments and explain what they changed (and/or did not change and why). Especially my queries regarding Figures 3, 4, 5A and B, exact abbreviations (e.g. line 263) remained un-answered. Also, my initial remark to cite the Kim publication and refer to the differences in the present work was half neglected (the claim that “this is the first report that L.c. affects adipogenesis and lipogenesis” has now been dropped).

I recommend a third review round with proper addressing the open queries, as otherwise I would recommend to reject the manuscript. 

Please accept our apologies for any carless mistake or error in the manuscript. Our manuscript has now been revised with all the required information, such as previously reported data on Leuconostoc spp on adipocyte differentiation and body weight reduction in response to Leuconostoc spp treatment, and other issues have been addressed. It is our hope that the reviewer will be satisfied with the revised manuscript.

  1. Kim et al., 2008, Lipid profile lowering effect of SoyproTMfermented with lactic acid bacteria isolated from Kimchi in high-fat diet-induced obese rats. BioFactors 33, 49-60. In that publication, a soy milk was fermented with lactic acid bacteria, among which L. c., and the effect of this product on plasma lipid levels and weight gain in rats and effect on adipocyte differentiation in pre-adipocyte 3T3-L1 cell lines. - Most likely, there is more literature existing on reporting similar effects in other fermented products. I would suggest that in the revised document, the authors cite this one reference and underline the difference of their study (e.g. cell-free supernatant of L.c. alone compared to a milk-based product that was produced using a combination of different lactic acid bacteria.

Thank you very much for your valuable suggestions. I have revised it in accordance with the recommendations of the reviewers. Currently, we have reviewed and included the data previously presented on Leuconostoc spp based fermented products and its biological application in different experimental models, along with the relevant references in the introduction, as well as discussed in the discussion section.

In introduction.

Several studies showing that Leuconostoc spp. mediated food supplements reduces obesity and metabolic diseases associated with obesity (Chu & Kim, 2018; Kim et al., 2008; Lee et al., 2018).   There has been a substantial variance in the present study in comparison to previously reported data on Leuconostoc species. We used a novel strain of Leuconostoc citreum in the present study to determine its efficiency in inhibiting differentiation and lipid accumulation in 3T3-L1 adipocytes. Therefore, the L. citreum strain was cultured in a 10% FBS-DMEM (Fetal Bovine Serum-Dulbecco's Modified Eagle Medium) medium for the production of secondary metabolites. The secondary metabolites in the sample were then lyophilized. The effects of the cell-free supernatant of L. citreum (LSC) on the differentiation of adipocytes and fat deposition have been studied. A further investigation was conducted on the molecular mechanisms that may underlie LSC's inhibition of lipogenesis and lipolysis.

In discussion

Several researchers have actively involved in production of fermented products in the presence of Leuconostoc spp with significant biological potential. As examples, the soymilk fermented with L. kimchi, L. citreum and L. plantarum significantly reduced fat deposition in 3T3-L1 adipocytes by inhibiting key transcription factors C/EBP-α and PPAR-γ. Furthermore, Soypro treatment reduced low density lipoprotein cholesterol (LDL) levels without affecting body weight in obese rats (Kim et al., 2008). Another study reported that pear extract and robusta fermented with L. mesenteroids significantly reduced body weight and adipose tissue mass. Also reduced the size of lipids in liver in obese rats compared to control rats (Choi et al., 2017; Chu & Kim, 2018). Supplementation with L. mesenteroids reduced blood urea nitrogen, glucose, and triglycerides levels in obese mice serum, as well as fatty liver development  and liver steatosis in comparison with controls (Castro-Rodríguez et al., 2020; Lee et al., 2018). Thus, we investigated effects of LSC on fat deposition and differentiation in 3T3-L1 adipocytes. But a significant difference exists between this study and previous studies on Leuconostoc species.

  1. Line 25, resistin … In contract, compared to control cells, adiponectin, … treated with LSC.

The resistin name has revised as the reviewer suggestion. Also, sentences have been revised as “But, compared to control cells, adiponectin, an insulin sensitizer, was elevated in adipocytes treated with LSC”.

  1. Line 35, … 10.7 % obese in their populations.

Now these lines have revised as Globally, obesity is becoming an epidemic issue, with an increasing occurrence rate. Obesity rate is 30.4% in the United States, 12.8% in Europe, and 10.7% in their populations in China

  1. Lien 39, According to Asper, .. What is Asper?

Now these lines have revised as According to OECD (Organization for Economic Co-operation & Development), more than 4% and almost 30% of the adult population in Korea are obese and overweight, respectively

  1. Line 41, Jung et al. 2020 (no first names in citations, please).

Yes, we strongly agreed with the reviewer comment and revised it same as Jung et al. 2020 projected that Korean adults would have a median body mass index (BMI) of 23.55 kg/m2 in 2040

  1. Line 47, … globally in context with obesity and related diseases.

Now it has revised as Every year, obesity and its related diseases kill more than 2.8 million people in worldwide

  1. Lines 65-66, the sentence seems to be fragmentary.

Now these sentences have been revised as “In the modern era, many drugs are available for treating obesity and its associated disorders. However, they can cause some adverse effects such nausea, insomnia, constipation, gastrointestinal problems, and cardiovascular problems”

  1. Line 66, please explain DMEM.

Preadipocytes of 3T3-L1 (ATCC-173, USA) were seeded into 96-well cell culture plates containing 10% fetal bovine serum in Dulbecco’s modified eagle medium (FBS-DMEM-30-2002)

  1. Line 72, … report showing that …

These sentences have been revised as “Several studies showing that Leuconostoc spp. mediated food supplements reduces obesity and metabolic diseases associated with obesity”

  1. Figure 1, why are the standard deviations not shown as error bars?

Sorry for this mistake. Now we have provided standard error for all results presented in the manuscript

  1. Line 82, citreum

Thanks for your valuable information. We have revised the name of the LAB in whole manuscript as citrum to citreum.

  1. Line 91, insulin, dexamethasone

The capital letter for Insulin and dexamethasone has be revised as per suggestion

  1. Line 95, strongly inhibited; well, ca. 30 % reduced compared to the control (Figure 2)

At 0.15 mg/mL, fat deposition was strongly inhibited compared to other dose ranges. LSC at 0.1 mg/mL also strongly decreased fat deposition in cells during differentiation

  1. Line 110, downregulates

Yes, revised as LSC downregulates differentiation and fatty acid synthesis associated proteins

  1. Line 114, CEB-a, should this be EBP-a?

Thanks for your information’s. Sorry. It is typographical error. Now it has been revised correctly

  1. Line 117, FAS and aP2 not shown in Figure 3B.

aP2 another name is FABP4. Now these lines have been revised as LSC treatment also inhibited the expression of key lipogenesis associated proteins such as FAS, ACC, and aP2. Other proteins related to insulin resistance and insulin sensitivity such as Resistin and adiponectin levels were also determined. Results showed that LSC treatment decreased Resistin but increased adiponectin expression in adipocytes than control (Figure 3A-B).

  1. Line 119, in adipocytes compared to …

Figure 3, A and B (WBs): why are there 3 controls left side and 2 controls right side? A, 0.05 mg/mL was the only concentration tested, why this one? What was the concentration in 3B?

First, we determined key adipocyte differentiation markers PPAR and CEBP beta and key lipogenic enzyme fatty acid synthase at multiple concentrations to find a suitable dose for further analysis. Therefore, we used the additional controls.

  1. Line 124, again, CEB-a or b, not shown in Figure 3, should this be EBP-a and b?

Now these sentences have been revised as “Western blot was used to determine molecular mechanisms involved in LSC treatment-induced fat reduction in adipocytes. Key transcriptional factors such as PPAR-γ, C/EBP-α/β, and SREBP-1c associated with differentiation were downregulated in adipocytes treated with LSC on day 10 (Figure 3A). LSC treatment also inhibited the expression of key lipogenesis associated proteins such as FAS, ACC, and aP2. Other proteins related to insulin resistance and insulin sensitivity such as resistin and adiponectin levels were also determined. Results showed that LSC treatment decreased resistin but increased adiponectin expression in adipocytes compared to control (Figure 3A-B)”.

  1. Figure 3 upper left side, what is CE?

Now it has revised as C/EBP

  1. Line 132, RGZ Line 136, RGZ treatment alone

Now it has revised as “RGZ treatment alone rapidly increased fat deposition”

  1. Figure 4C, why two WB samples per molecule tested? Rosi should be RGZ. What shall the bands for GAPDH indicate?

Rosi has been revised as RGZ. Here we used two independent replicate samples were used to confirm the expression of master protein PPAR expressions. GAPDH is a house keeping gene can be used to calculate fold of target protein expression after normalization with GAPDH.

  1. Figure 5A, why again two samples for control and for LSC? Why are the results described and shown in the column chart do not fit the WB results (the bands in the WB seem to be of almost identical intensity)?

There is no identical intensity between groups. After normalized with housekeeping gene GAPDH, this result showed some significant changes in targeted protein expression. Here we used two independent replicate samples were used to confirm the expression of master protein PPAR expressions. GAPDH is a house keeping gene can be used to calculate fold of target protein expression after normalization with GAPDH.

  1. Line 158, pp38 MAPK or p38 MAPK? pERK 44/42 or Erk ½?

pp38 MAPK. It indicates phosphorylated status of p38.Eê°€ 44/42 and Eê°€ 1/2 are the same.

  1. Line 167, AICAR?

Now explanation for AICAR has been provided in respective place. “Differentiated adipocytes treated with LSC at 0.15 mg/mL or AICAR (5-Aminoimidazole-4-carboxamide ribonucleotide) at 1 mM, an AMPK-α agonist, for 12 h increased phosphorylation of AMPK-α closely associated with lipolysis but decreased PPAR-γ, FAS, and ACC expression levels closely associated with adipocyte differentiation and fatty acid synthesis (Figure 5). Results obtained for LSC were significantly comparable with AICAR compared to control cells”.

  1. Why Figure 5A and 5B and not 5 and 6?

Because both figures are relevant to molecular signaling involved in the adipocyte differentiation and lipid accumulation. However, Now the figure number has been revised as figure 5 and figure 6 according to reviewer suggestion.

  1. Figure 5B right part, column chart, x-axis, shouldn’t it be p(p)MAPK and not AMPK?

Figure 5B now it is figure 6, I could not understand the reviewer comment on x axis p(p)MAPK and not AMPK? We have mentioned correctly as phospho-AMPK as pAMPKα in both graph and image.

Figure 5A now it is figure 5, we analysed the status of pp38MAPK and mentioned as it in both graph and image.Now we have revised all figures labelling according to the reviewer suggestion.

  1. Line 211, LSC and RGZ seem to counteract, so there’s almost no difference between control and LSC+RGZ as shown in Figure 4C.

Yes, definitely, there is no significant difference between control and LSC+RGZ treatments. But, compared RGZ treatment, LSC+RGZ treatment had significant reduction at 0.05level.

  1. Line 211, whereas

The space was deleted

  1. Line 238, in contrast,

It has been revised as However, a high level of resistin is associated with insulin resistance and diabetes

  1. Line 262, activates

Yes, it has revised

  1. Line 263, PPAR and ACC effect, again, the WB results look different (comparable to same intensities) compared to the column chart on the right side of the Figure.

Yes, definitely, there is no significant difference between control and LSC+RGZ treatments. But, compared RGZ treatment, LSC+RGZ treatment had significant reduction at 0.05level.

  1. Line 270, was comparable

Yes, revised as by increasing its phosphorylating level. Result after LSC treatment was comparable to those after AICAR treatment

  1. Line 279, followed by lyophilisation. How long was sample lyophilized? 50 mTorr (about 0.07 mbar)

Now it has revised as filtrating with filter membranes having different pour sizes, and then lyophilized at -40 °C under less than 50 m Torr pressure for 72h

Line 70-71. These sentences have been revised as Therefore, in the present study, we used a novel strain of Leuconostoc citreum to produce secondary metabolites (LSC) in DMEM (Dulbecco's Modified Eagle Medium).  The secondary metabolites in sample was then lyophilized 70-71

  1. Line 287, Then microliters of

These lines have been revised as EZ-cytox reagent (DoGenBio, Seoul Korea) (10 µL/well) was added to each well followed by incubation at 37 °C with 5% CO2 for a further 30 min

  1. Line 318, semi-dry

Now revised as Afterwards, proteins were transferred to polyvinylidene difluoride membranes (PVDF) using a semiwet transfer method

  1. Line 335, enhanced lipolysis? Reduced it should be.

Thank you for your kind information and revised as Cells treated with LSC for 12 h showed enhanced lipolysis by increasing phosphorylation of AMPK-α at Thr172 and inhibiting lipogenesis-associated enzymes FAS and ACC. It is correct hypothesis. In general, increases of lipolysis means breakdown of lipid has been increased. So, it is an essential to stimulate the lipolysis to reduce body fat content.

  1. Line 338, a very effective

The sentences have been revised as Overall, results of this study suggest that LSC has potential as very effective multifunctional probiotic bacteria with anti-obesity activity

  1. References Please use journal abbreviations

We have used endnote software with MDPI endnote style for reference preparations and again revised it manually.

Castro-Rodríguez, D. C., Reyes-Castro, L. A., Vega, C. C., Rodríguez-González, G. L., Yáñez-Fernández, J., & Zambrano, E. (2020). Leuconostoc mesenteroides subsp. mesenteroides SD23 Prevents Metabolic Dysfunction Associated with High-Fat Diet–Induced Obesity in Male Mice. Probiotics and Antimicrobial Proteins, 12(2), 505-516. doi:10.1007/s12602-019-09556-3

Choi, S. Y., Ryu, S. H., Park, J. I., Jeong, E. S., Park, J. H., Ham, S. H., . . . Choi, Y. K. (2017). Anti-obesity effect of robusta fermented with Leuconostoc mesenteroides in high-fat diet-induced obese mice. Exp Ther Med, 14(4), 3761-3767. doi:10.3892/etm.2017.4990

Chu, H., & Kim, J. (2018). Anti-Obesity Effect of Fructus Pyri Pyrifoliae Extract Fermented by Lactic-Acid Bacteria on Rats. Applied Microscopy, 48, 62-72. doi:10.9729/AM.2018.48.3.62

Kim, N. H., Moon, P. D., Kim, S. J., Choi, I. Y., An, H. J., Myung, N. Y., . . . Kim, H. M. (2008). Lipid profile lowering effect of Soypro fermented with lactic acid bacteria isolated from Kimchi in high-fat diet-induced obese rats. Biofactors, 33(1), 49-60. doi:10.1002/biof.5520330105

Lee, S. Y., Sekhon, S. S., Ko, J. H., Kim, H. C., Kim, S. Y., Won, K., . . . Kim, Y.-H. (2018). Lactic Acid Bacteria Isolated from Kimchi to Evaluate Anti-obesity Effect in High Fat Diet-induced Obese Mice. Toxicology and Environmental Health Sciences, 10, 11-16.

Round 3

Reviewer 5 Report

The authors have carefully addressed my remaining comments and have revised the manuscript accordingly. It is now accepted for publication.